# Membrane-Bound EMC10 Is Required for Sperm Motility via Maintaining the Homeostasis of Cytoplasm Sodium in Sperm

**DOI:** 10.3390/ijms231710069

**Published:** 2022-09-03

**Authors:** Lijie Liu, Shanhua Mao, Kuangyang Chen, Jiarong Dai, Shuoshuo Jin, Lijiao Chen, Yahao Wang, Lina Guo, Yiting Yang, Chongwen Zhan, Zuquan Xiong, Hua Diao, Yuchuan Zhou, Qiang Ding, Xuanchun Wang

**Affiliations:** 1Department of Urology, Huashan Hospital, Fudan University, Shanghai 200040, China; 2Department of Endocrinology, Huashan Hospital, Fudan University, Shanghai 200040, China; 3NHC Key Laboratory of Reproduction Regulation (Shanghai Institute for Biomedical and Pharmaceutical Technologies), School of Basic Medical Sciences, Fudan University, Shanghai 200032, China; 4Department of General Surgery, Huashan Hospital, Fudan University, Shanghai 200040, China; 5Shanghai Key Laboratory of Embryo Original Diseases, The International Peace Maternity and Child Health Hospital, School of Medicine, Shanghai Jiao Tong University, Shanghai 200030, China

**Keywords:** membrane-bound EMC10, secreted EMC10, ATP1B3, *Emc10* knockout mouse, cytoplasm sodium, sperm motility

## Abstract

Endoplasmic reticulum membrane protein complex subunit 10 (EMC10) is an evolutionarily conserved and multifunctional factor across species. We previously reported that *Emc10* knockout (KO) leads to mouse male infertility. *Emc10*-null spermatozoa exhibit multiple aspects of dysfunction, including reduced sperm motility. Two subunits of a Na/K-ATPase, ATP1A4 and ATP1B3, are nearly absent in *Emc10* KO spermatozoa. Here, two isoforms of EMC10 were characterized in the mouse testis and epididymis: the membrane-bound (mEMC10) and secreted (scEMC10) isoforms. We present evidence that mEMC10, rather than scEMC10, is required for cytoplasm sodium homeostasis by positively regulating ATP1B3 expression in germ cells. Intra-testis mEMC10 overexpression rescued the sperm motility defect caused by *Emc10* KO, while exogenous recombinant scEMC10 protein could not improve the motility of spermatozoa from either *Emc10* KO mouse or asthenospermic subjects. Clinically, there is a positive association between ATP1B3 and EMC10 protein levels in human spermatozoa, whereas no correlation was proven between seminal plasma scEMC10 levels and sperm motility. These results highlight the important role of the membrane-bound EMC10 isoform in maintaining cytoplasm sodium homeostasis and sperm motility. Based on the present results, the mEMC10-Na, K/ATPase α4β3 axis is proposed as a novel mechanism underlying the regulation of cytoplasmic sodium and sperm motility, and its components seem to have therapeutic potential for asthenospermia.

## 1. Introduction

Infertility refers to the incapacity to become successfully pregnant following more than one year of normal sexual life without any contraceptive measures, probably due to reproductive system disorders in men or women [1]. Clinical studies have shown that about 15–25% of couples struggle with fertility issues [2,3], of which male factors account for approximately 50% [1]. Male infertility is attributed to various causes [4], with genetic mutations accounting for over 15% of all cases [5]. Previously, we identified a novel protein that plays a crucial role in male fertility: endoplasmic reticulum membrane protein complex 10 (EMC10) [6].

Endoplasmic reticulum membrane protein complex (EMC) is an evolutionarily conserved and multifunctional protein complex, containing 10 subunits (i.e., EMC1–EMC10) in mammals [7]. Human EMC10 consists of two isoforms: secreted EMC10 (scEMC10) and membrane-bound EMC10 (mEMC10) [8]. scEMC10 is detectable in both human and mouse sera [9,10]. Structural biology assays show that mEMC10 serves as a lumenal subunit of EMC anchored in the endoplasmic reticulum membrane [7,11]. In previous studies, we first cloned human scEMC10, then designated as INM02, from a complementary DNA library of human insulinoma tissue and revealed its potential roles in the regulation of pancreatic beta cell function and glucose metabolism [9,12]. Subsequent studies have indicated that scEMC10 exerts an inhibitory role in cell proliferation, migration, and invasion of human glioma [8,13], and a sustained beneficial effect on tissue repair after myocardial infarction via promoting angiogenesis in mice [10], as well as the unique function of mEMC10 in rescuing the schizophrenia-related deficits [14,15]. Recently, several genetic studies indicated that either a frameshift variant or a single-nucleotide polymorphism (SNP) in *EMC10* is associated with a new syndrome with intellectual disability and language impairment in humans [16,17,18].

In a previous study, we successfully established an *Emc10* knockout (KO) mouse model and observed that EMC10 deficiency resulted in male infertility [6]. Although causing no significant decrease in sperm counts, *EMC10* KO causes several sperm dysfunctions, including decreased motility, abnormal morphology, impaired capacitation, and impotent acrosome reaction [6]. Several mechanisms have been proposed for the male sterility caused by *Emc10* KO. The ablation of EMC10 leads to the inactivation of a sodium/potassium (Na/K)-ATPase, evidenced by dramatic reductions in the protein levels of its subunits, ATP1A4 and ATP1B3, in spermatozoa, therefore causing increased intracellular sodium concentration, which accounts for decreased motility and abnormal morphology of *Emc10* KO spermatozoa. The absence of EMC10 results in a decrease in HCO3^−^-induced pH and subsequent reductions in both cyclic adenosine monophosphate (cAMP)-dependent protein kinase (PKA) substrate phosphorylation and protein tyrosine phosphorylation, contributing to the impaired capacitation of *Emc10*-null spermatozoa [6].

In this study, we aimed to distinguish which isoform of EMC10 is involved in the regulation of ATP1B3 expression and cytoplasm sodium in sperm cells, and investigate their therapeutic value in *Emc10* KO mice. In this context, we identified mEMC10 to be necessary for sperm motility, as well as in ATP1B3 expression and cytoplasm sodium regulation. Furthermore, we explored whether scEMC10 can serve as a biomarker for sperm motility and its therapeutical potential for asthenozoospermia in humans.

## 2. Results

### 2.1. Cloning, Protein Expression Profile and Glycosylation Analysis of Mouse EMC10

To date, two isoforms of *EMC10* have been reported in humans, but no such information is available regarding the mice *EMC10* gene [8]. Two products were also amplified from the testis and epididymal tissues of mouse using a single pair of primers targeting the mouse *Emc10* gene (Figure 1a). DNA sequencing analysis revealed that the larger product consisted of an open reading frame (ORF) of 975 bp encoding a protein of 264 aa, which corresponded to mouse *Emc10* transcript variant 2 in the GenBank database, while the smaller one contained an ORF of 777 bp encoding a 258 aa protein, linked to mouse *Emc10* transcript variant 1 (Figure 1a and Appendix A). The two isoforms of mouse EMC10 share a homology of 227 aa (Appendix A). The variant splicing of mouse *Emc10* gene gives rise to a transmembrane domain in the C terminal of EMC10-1, and a signal peptide domain in the N terminal of EMC10-2 (Appendix A).

When we explored the EMC10 protein expression in the testis and epididymis tissues of mouse, two bands with different molecular weights were observed in sodium dodecyl-sulfate polyacrylamide gel electrophoresis (SDS-PAGE) gels using an anti-EMC10 polyclonal antibody (Figure 1b). Theoretically, the six aa between the two isoforms is unlikely to result in such a big separation in the gel. Glycosylation is the most common protein modification in mammals, which can alter protein structure and molecular weight [19]. It has been shown that the human scEMC10 bears a complex glycosylation [13]. Thus, we hypothesized that mouse EMC10 is also glycosylated. We observed that *N*-glycosidase F treatment slightly decreased the sizes of both bands (Figure 1b), whereas *O*-glycosidase cleavage remarkably changed the running pattern of mouse EMC10 protein with a disappearance of the larger band (Figure 1c), suggesting that mouse EMC10 indeed undergoes glycosylation, especially *O*-linked glycosylation.

Our previous evidence has shown that rat *Emc10* gene is expressed in multiple tissues [9]. Here, we observed that the EMC10 protein was ubiquitously distributed in mouse tissues with high levels in liver, testis, and epididymis (Figure 1d). Furthermore, immunohistochemistry (IHC) analysis showed that EMC10 was predominately expressed in spermatogonia, and sperm EMC10 levels gradually increase in spermatozoa from caput to cauda epididymis, suggesting a role of EMC10 in sperm maturation (Figure 1e).

### 2.2. Membrane-Bound EMC10 Maintains the Homeostasis of Cytoplasm Sodium via Positively Regulating ATP1B3 in Mouse Germ Cells

ATP1B3, a member of the Na/K-ATPase beta subunit family, has not been linked to male fertility. We firstly investigated whether ATP1B3 is expressed in spermatozoa. IHC staining indicated that ATP1B3 is present at a low level in the spermatozoa of testis, and at a high level in the spermatozoa of corpus and cauda epididymis, which was further supported by the Western blotting results showing much higher levels of ATP1B3 in the spermatozoa than in the testis tissues (Figure 2a,b). In mouse germ cells GC-1 spg and GC-2 spd, ATP1B3 and EMC10 were also confirmed to be highly expressed (Figure 2b). GC-1 spg, which is representative of the stage between type B spermatogonia and preleptotene spermatocytes, was arrested at an early stage of meiosis [20]. GC-2 spd was derived from 6-week-old mouse testis and represented the stage between preleptotene spermatocytes and round spermatids [21]. Although several other germ cells have been established, only GC-1 spg and GC-2 spd are commercially available at present. In addition, immunofluorescence staining, which was performed to localize ATP1B3 in mouse spermatozoa, revealed that ATP1B3 was confined to the midpiece of spermatozoa from epididymis regardless of caput, corpus, or cauda regions (Figure 2c); negative controls are depicted in Appendix A. Previously, we localized EMC10 in the principal piece of testicular spermatozoa, and found it delocalized from the midpiece to acrosome during epididymal transit [6]. The spatial overlap of ATP1B3 and EMC10 in the midpiece of spermatozoa suggested a potential interaction between them.

In order to explore the role of EMC10 in the regulation of cytoplasm sodium and ATP1B3 expression, we firstly silenced EMC10 expression in both GC-1 and GC-2 germ cells using small interference RNA (siRNA). The results indicated that knockdown of EMC10 significantly decreased both the protein and mRNA levels of ATP1B3 in the germ cells (Figure 3a,b). We previously observed much higher intracellular sodium levels in *Emc10* KO spermatozoa than in wild-type spermatozoa [6]. As predicted, knockdown of EMC10 also significantly increased cytoplasm sodium concentration in both GC-1 and GC-2 cells (Figure 3c). Due to the high homology in cDNA sequences between *scEmc10* and *mEmc10* gene, the siRNA used here silenced the expression of both *scEmc10* and *mEmc10*. To distinguish mEMC10 from scEMC10, a plasmid-encoding mEMC10 was transfected into germ cells, and ATP1B3 expression and cytoplasm sodium were determined. It was found that overexpression of mEMC10 results in significant increases in both ATP1B3 protein and mRNA levels in the two germ cells (Figure 3d,e), associated with significant decreases in cytoplasm sodium (Figure 3f), which are contrary to the findings obtained from the siRNA-mediated knockdown experiments. In another way, the incubation of exogenous recombinant scEMC10 protein exerted no effects on either ATP1B3 expression or cytoplasm sodium concentration (Appendix A). These data clarify that mEMC10, but not scEMC10, is involved in the regulation of ATP1B3 expression and cytoplasm sodium in germ cells.

To determine whether ATP1B3 is involved in the mEMC10 regulation of cytoplasm sodium, siRNA-mediated knockdown of ATP1B3 in mEMC10-overexpressed germ cells was performed and cytoplasm sodium was measured. ATP1B3 siRNA efficiently reduced both protein and mRNA levels of ATP1B3 in mEMC10-overexpressed or control germ cells (Figure 3g,h). In cells transfected with a control plasmid, ATP1B3 silencing significantly increased cytoplasm sodium (Figure 3i). Notably, knockdown of ATP1B3 significantly blunted the impacts of mEMC10 on the regulation of cytoplasm sodium (Figure 3i), implying that the mEMC10-regulating cytoplasm sodium is dependent on ATP1B3.

### 2.3. Intra-Testis Overexpression of Membrane-Bound EMC10 Improves Sperm Motility in Emc10 KO Mice

We then explored whether male infertility could be rescued by reintroduction of either mEMC10 or scEMC10 in vivo. First, intra-testis overexpression of mEMC10 in *Emc10* KO mouse was achieved via the injection of lentivirus expressing mEMC10 into the rete testis (Figure 4a). One month after injection, sperm motion parameters were analyzed using computer-assisted sperm analysis (CASA). Visually, the green movement trajectories were increased dramatically in mEMC10-overexpressed spermatozoa (Figure 4b). Furthermore, CASA demonstrated that multiple parameters of sperm motion were significantly higher in mEMC10-overexpressed KO spermatozoa than in controls, including the percentages of total motility and progressive motility, and the velocities of average path (VAP), curvilinear (VCL), and straight line (VSL), except beat cross frequency (BCF), which are near to the ones of wild-type spermatozoa (Figure 4c). These data indicate that the restoration of mEMC10 in testis could rescue the motile defects caused by *Emc10* KO and suggest a therapeutical effect of mEMC10 on asthenozoospermia.

Next, scEMC10 was reintroduced into the circulation of *Emc10* KO mouse via tail-vein injection of an adeno-associated virus serotype 8 expressing scEMC10 (AAV8-scEMC10) (Appendix A). No more movement trajectory was observed after the reintroduction of scEMC10 (Appendix A). In addition, there were no significant differences in all parameters associated with sperm motility between AAV8-scEMC10 mice and LacZ control, including percentages of total motility and progressive motility, VAP, VSL, VCL, or BCF (Appendix A). To exclude the possibility that the negative effects were caused by circulatory scEMC10 not penetrating the blood–testis barrier, we treated *Emc10* KO spermatozoa with exogenous recombinant scEMC10 in vitro. In agreement with the in vivo study, the in vitro incubation of scEMC10 was not capable of improving any parameter of sperm motility (Appendix A).

### 2.4. The Levels of ATP1B3 Protein Are Positively Correlated with Those of EMC10 Protein in Human Spermatozoa

We found that sperm EMC10 levels are positively correlated with sperm motility in humans [6]. In the present study, another cohort of 61 human semen samples was gathered and the levels of both EMC10 and ATP1B3 were determined in sperm cells. Western blot analysis revealed individual variations in the levels of both EMC10 and ATP1B3 in spermatozoa (Figure 5a–e). Correlation analysis indicated that the levels of ATP1B3 were positively associated with those of EMC10 in human spermatozoa (Figure 5f). To a certain extent, this finding supports the observation that mEMC10 positively regulates ATP1B3 in germ cells (Figure 3). Currently, the antibody used to immunoblot EMC10 could not differentiate mEMC10 from scEMC10. An antibody specifically recognizing mEMC10 will clarify which isoform of EMC10 is positively correlated with ATP1B3 in human spermatozoa.

### 2.5. Seminal Plasma scEMC10 Levels Are Not Associated with Sperm Motility in Humans

The scEMC10 isoform is present in both human and mouse sera [9,10]. Here, a cohort of 189 Chinese participants was recruited and categorized into normal-motile (total motility ≥40% and progressive motility ≥32%) and low-motile spermatozoa (total motility <40% or progressive motility <32%) groups according to motility-associated seminal parameters. scEMC10 in seminal plasma of these participants was examined using a chemiluminescence immunoassay (CLIA) kit developed at our lab. There were no significant differences in seminal plasma scEMC10 levels between the normal-motile and the low-motile groups (Table 1).

To check whether there is a non-linear correlation between seminal plasma scEMC10 levels and sperm motility, the participants were grouped into scEMC10 quartiles. No significant differences were observed in the various sperm parameters among scEMC10 quartiles (Table 2). Furthermore, Spearman’s correlation analysis indicated no significant correlations between seminal plasma scEMC10 levels and various parameters including sperm count, sperm motility, PR, VAP, VSL, VCL, and BCF (Table 3). Taken together, seminal plasma scEMC10 levels were not associated with sperm motility in humans, suggesting that seminal plasma scEMC10 was not a superior biomarker for evaluating sperm motility and male fertility.

### 2.6. scEMC10 Is Not Capable of Improving the Motility of Asthenospermic Sperm

Finally, to determine whether scEMC10 has therapeutic potential for asthenospermia, 36 sperm samples from patients diagnosed with asthenospermia were incubated with recombinant scEMC10 protein in vitro. No significant improvements in any parameters linked to sperm motility were observed in the scEMC10-treated group compared to the control group (Figure 6a,b). Together with the finding that in vitro incubation of recombinant scEMC10 could not improve motility of *Emc10* KO spermatozoa, these results suggest that recombinant scEMC10 has limited therapeutic value for patients with asthenospermia. Whether it can treat male infertility with other etiologies will be determined in future studies.

## 3. Discussion

In this study, we differentiated for the first time a special role of the membrane-bound EMC10 rather than the secreted EMC10 in the regulation of cytoplasm sodium and sperm motility. Furthermore, we observed a positive correlation between EMC10 and ATP1B3 in human sperm, while no application value was confirmed for scEMC10 in the diagnosis and treatment of asthenospermia.

As reported in humans, scEMC10 is a new class of secreted factor and can be detected in blood circulation [8,9]. Nevertheless, mEMC10 and other EMC proteins form a complex anchored to the endoplasmic reticulum membrane, where mEMC10 is localized in the lumen of the endoplasmic reticulum [7]. Their distinct localizations presage the different functions of the two isoforms, which have been validated by several lines of evidence [8,9,10,13,14,15]. Here, two isoforms of the *Emc10* gene were characterized in the testis and epididymis of mice. DNA sequencing and bioinformatics analysis identified the two isoforms as the Emc10 transcript variants 1 and 2 deposited in GenBank, corresponding to mEmc10 and scEmc10, respectively. Meanwhile, mEMC10 was proved to be the exact isoform that functions in the regulation of cytoplasm sodium and sperm motility.

Intracellular ion balance, specifically Na^+^, is of critical importance for maintaining sperm motility [22]. The Na/K-ATPase, consisting of two major polypeptides, the α and β subunits, utilizes the energy from the hydrolysis of ATP to exchange cytoplasmic Na^+^ for extracellular K^+^, thus maintaining the intracellular sodium concentration at a low level [23]. Proteomic profiling, together with Western blotting assays, revealed almost absences of two subunits of a Na/K-ATPase, ATP1A4 and ATP1B3, in *Emc10* KO spermatozoa, suggesting a regulatory role of EMC10 in the activity of the Na/K-ATPase [6]. In this study, we presented direct evidence that knockdown of EMC10 enhances intracellular sodium concentration in germ cells, whereas overexpression of mEMC10 reduces it. In addition, we determined that the decreased cytoplasm sodium caused by mEMC10 overexpression is significantly blunted by ATP1B3 knockdown, indicative of ATP1B3 as a connector between mEMC10 and cytoplasm sodium of sperm. It has been established that the subunit α4, predominating in male germ cells, can associate with the subunit β3 to render a competent Na/K-ATPase, α4β3 [24]. Collectively, we propose a novel molecular mechanism by which mEMC10 maintains the low intracellular sodium concentration of sperm via positively regulating the Na/K-ATPase α4β3. We previously showed that EMC10 is localized to the midpiece of spermatozoa from the caput and corpus of the epididymis [6]. ATP1A4 has also been reported to be present in the central region of the flagellum of spermatozoa from caput and cauda of the epididymis [25]. In this study, the immunofluorescence data indicated that ATP1B3 is confined to the midpiece of spermatozoa from the epididymis. The spatial overlap of these three components also supports their synergistic effect.

It has been described that ion balance and low intracellular sodium concentration are required for normal sperm motility [26]. We observed that *Emc10*-null spermatozoa have significantly lower percentages of motility and decreased velocities of path, linear, and track, accompanying higher intracellular sodium concentration, when compared with wild-type spermatozoa [6]. ATP1A4 has also been indicated to play a crucial role in maintaining cytoplasm sodium concentration and sperm motility. Its deficiency contributes to male infertility and multiple sperm dysfunctions, including low sperm motility and high intracellular sodium concentration, similar to the phenotype of *Emc10* KO mouse [27,28,29], while overexpression of ATP1A4 in transgenic mice significantly increased sperm motility [30]. Here, we present evidence that intra-testis overexpression of mEMC10 rescues the motility defects of *Emc10*-null spermatozoa, and in vitro overexpression of mEMC10 increases the expression of ATP1B3 and reversely decreases the intracellular sodium concentration in germ cells. These findings highlight a crucial role for mEMC10 in governing intracellular sodium concentration and sperm motility, which is fulfilled via its positive impact on the activity of the Na/K-ATPase, α4β3. However, the impotent effects of scEMC10 on sperm motility coincide with the in vitro observations where neither ATP1B3 expression nor cytoplasm sodium in germ cells were altered by scEMC10 incubation. Taken together, our present results suggest that the mEMC10-Na, K/ATPase α4β3 axis plays a fundamental role in maintaining low intracellular sodium concentration and sperm motility. Clinically, both EMC10 and ATP1A4 protein levels are positively associated with sperm motility in humans [6], and a positive correlation was also observed between EMC10 and ATP1B3 protein levels in human spermatozoa in this study, implying that the components in the axis exhibit diagnostic and therapeutic potential for asthenospermia or male infertility.

Currently, semen analysis is known to be the gold standard for the clinical diagnosis of male infertility [31,32]. Reactive oxygen species (ROS) and sperm DNA fragmentation (SDF) have also been explored as strategies for the assessment of sperm quality and prediction of male fertility, but the results obtained are unsatisfactory [33,34]. Although no effects of scEMC10 on sperm motility were observed, we still explored whether scEMC10 might be a biomarker for sperm motility or male fertility. Unfortunately, there were no differences in seminal plasma EMC10 levels between subjects with normal and low-motile spermatozoa, nor was there a correlation between seminal plasma EMC10 and sperm motility. In another cohort, a positive correlation was observed between EMC10 and ATP1B3 protein levels in human spermatozoa. Considering that ATP1B3 is regulated by mEMC10, the components in the mEMC10-Na, K/ATPase α4β3 axis likely serve as biomarkers for the evaluation of sperm motility or male fertility.

Despite the progress achieved in this study, some uncertainties and limitations remain. First, the specific antibody that can distinguish mEMC10 from scEMC10 should be developed to further clarify the role of EMC10 in male reproduction. The strategy of intra-testis overexpression of mEMC10 did not render female mouse conceived, nor did they produce offspring, in spite of the increased sperm motility. Besides the defect of motility, *Emc10*-null spermatozoa also exhibited abnormal morphology, impaired capacitation, and impotent acrosome reaction; whether these could be rescued by mEMC10 has not been determined. In addition, we showed that ATP1B3 expression is positively controlled by mEMC10 and is required for the mEMC10-regulation of cytoplasm sodium. Whether ATP1A4, another subunit of the Na/K-ATPase α4β3 that is also absent in *Emc10*-null sperm, is involved in the regulation of intracellular sodium concentration will also be explored in the future.

In conclusion, this work represents the first effort to explore the impacts of the distinct isoforms of EMC10 on a certain biological activity and reveals that mEMC10, rather than scEMC10, governs the low intracellular sodium concentration in sperm via positively regulating ATP1B3 and is required for maintaining the sperm motility. Based on this study, the mEMC10-Na, K/ATPase α4β3 axis is proposed as a novel mechanism underlying the regulation of cytoplasm sodium and motility of sperm, and its components emerge as therapeutic potentials for asthenospermia.

## 4. Materials and Methods

### 4.1. Patients and Semen Samples

Semen samples (*n* = 189) were collected at the Huashan Hospital (Shanghai, China), between July 2021 and November 2021. Based on the analysis of sperm motility and progressive motility, 67 patients were diagnosed with asthenospermia, while the remaining 122 had normal sperm motility. The study protocol was approved by the Ethics Committee of Huashan Hospital (protocol number KY2021-767, 17 August 2021). Furthermore, this study was carried out in accordance with the Declaration of Helsinki, and written informed consent was obtained from all participants.

To isolate sperm cells from semen samples, 50% Percoll (17089109, GE Healthcare, Sweden) centrifugation was performed at 500 g for 15 min at 37 °C. Seminal plasma was extracted as previously described [35], and the content of scEMC10 in seminal plasma was detected by using CLIA. scEMC10 recombinant protein was obtained from Shanghai Kehua Bioengineering Co., Ltd. (Shanghai, China), and its biological activity was verified as previously described [9]. After dilution with pre-equilibrated Bigger–Whitten–Whittingham (1 × BWW) to a concentration of 10~20 × 10^6^ cells per mL, the sperm samples were incubated with 1 μg/mL scEMC10 recombinant protein at 37 °C for 1 h, followed by the detection of sperm motion parameters using CASA (IVOS Ⅱ, Hamilton Thorne, Beverly, MA, USA).

### 4.2. Measurement of scEMC10 in Seminal Plasma by CLIA

The scEMC10 luminescent immunoassay kit was procured from Beijing North Institute of Biotechnology (Beijing, China), and the detection method was as described previously [9]. First, 50 μL of scEMC10 standards or seminal plasma samples was added to the corresponding wells of the immunoplate. The same volume of scEMC10 monoclonal antibody IF12 horseradish peroxidase (HRP) conjugate was then dispensed into each well, followed by immunoplate sealing with an acetate plate sealer. After incubation at 4 °C overnight, each well was aspirated and then washed 4 times with 350 μL of 1× washing buffer. Subsequently, 100 μL of substrate solution was added to each well for incubation at 37 °C for 5–8 min, avoiding light exposure. Finally, the immunoplate was loaded onto a chemiluminescent microplate reader and the relative luminescence units (RLUs) of each well were measured at 360–470 nm. Particularly, the concentrations of scEMC10 standards were 0, 0.3, 1.5, 7.5, 30, and 120 ng/mL.

### 4.3. Animal Experiments

C57BL/6 mice were obtained from Jiangsu Gempharmatech Co., Ltd. (Nanjing, China) and *Emc10* KO mice were established as previously described [6]. They were housed in a barrier animal room with a normal light/dark cycle and could freely access food and water. All the experimental procedures were performed according to the experimental standards of Huashan Hospital and international guidelines on animal experiments. The study protocol was approved by the Ethics Committee of Huashan Hospital (Shanghai, China).

To rescue sperm motility and fertility of *Emc10* KO male mice, we prepared concentrated lentivirus (2.16 × 10^8^ TU/mL; Genomeditech, Shanghai, China) and adeno-associated virus (2.5 × 10^11^ vg/mL; Viral Core, Boston Children Hospital, USA) to overexpress mEMC10 and scEMC10, respectively. First, we injected the mEMC10 lentivirus or unloaded lentivirus into the rete testes of 6-week-old *Emc10* KO mice. One month later, mEMC10 overexpression efficiency in testis was verified via immunohistochemistry. We also injected scEMC10 virus or LacZ virus through the tail vein of 6-week-old *Emc10* KO mice. One month later, scEMC10 expression in the serum was measured using Western blotting. In addition, we observed whether offspring were born to male mice overexpressing mEMC10 and scEMC10 one month after injection. Sperm from the caudal epididymis was obtained and sperm motion parameters were detected using CASA (IVOS II, Hamilton Thorne, Beverly, MA, USA).

In vitro, sperm from the caudal epididymis of *Emc10* KO male mice was incubated with 1 μg/mL scEMC10 recombinant protein at 37 °C for 1 h. Thereafter, sperm motion parameters were analyzed using CASA (IVOS II, Hamilton Thorne, Beverly, MA, USA).

### 4.4. Cell Culture

The mouse male germ cell line GC-1 spg was bought from American Type Culture Collection (CRL-2053; Manassas, VA, USA) and GC-2 spd was from the National Collection of Authenticated Cell Cultures (SCSP-5055; Shanghai, China). Both cells were cultivated in Dulbecco’s modified eagle medium (Bioind, Israel) supplemented with 10% fetal bovine serum (Gibco, Thermo Fisher Scientific, Carlsbad, CA, USA) and antibiotics (1% streptomycin/penicillin, Gibco, USA). Cell culturing was performed at 37 °C with 5% CO_2_. The scEMC10 recombinant protein was diluted to different concentrations (0, 100, and 1000 ng/mL) for cell incubation.

### 4.5. Construction of Stably Transfected Cells

The mEMC10-overexpressed plasmid ligated into the puromycin vector was purchased from Genomeditech (Shanghai, China). After packaging by lentivirus, the plasmid was transfected into the GC-1 and GC-2 cells. Briefly, both cells were cultured in 24-well plates until 30–50% confluence. Then, the lentivirus (2.16 × 10^8^ TU/mL; Genomeditech, Shanghai, China) was added into the culture medium to infect the cells for 16 h. To obtain the stable cells, clones overexpressing mEMC10 were selected using 4 μg/mL puromycin (A1113803, Gibco, Thermo Fisher Scientific, Carlsbad, CA, USA) for two weeks.

### 4.6. RNA Interference

Small interfering RNAs (siRNAs) were purchased from Genomeditech (Shanghai, China). The RNA interference sequences were listed as follows: EMC10 siRNA, 5′-AGUCUUUCUUUGCCAAAUA-3′; ATP1B3 siRNA, 5′-CCAGACGGAUAUCCACAAAUA-3′. The siRNAs were transiently transfected into GC-1 and GC-2 cells using Lipofectamine 3000 (L3000150, Invitrogen, Carlsbad, CA, USA) following the manufacturer′s instructions. Specially, EMC10 siRNA can silence mEMC10 and scEMC10 simultaneously.

### 4.7. Western Blotting Analysis

Germ cells, mouse tissues, and human spermatozoa were lysed in radioimmunoprecipitation assay buffer (P0013B, Beyotime, Shanghai, China), and protein concentration analysis was performed using a bicinchoninic acid (BCA) protein assay kit (23225, Invitrogen, Carlsbad, CA, USA). First, equal quantities of proteins were separated on 12% (*w*/*v*) SDS-PAGE and placed on a polyvinylidene fluoride (PVDF) membrane activated by methanol, followed by incubation with blocking solution (5% milk) at 37 °C for 1 h. The PVDF membranes were then incubated with antibodies against EMC10 (1:2000), ATP1B3 (1:5000; ab231671, Abcam, Cambridge, MA, USA), α-tubulin (1:2000; AF2827, Beyotime, Shanghai, China), and β-actin (1:10,000; ab6276, Abcam, Cambridge, MA, USA) at 4 °C overnight. EMC10 polyclonal antibody was generated as described previously [9]. After washing 3 times with 1× tris-buffered saline and Tween (TBST), the membranes were incubated with secondary antibodies against rabbit or mouse (1:1000; A0208 or A0216, Beyotime, Shanghai, China) at 37 °C for 1 h, prior to enhanced chemiluminescence detection (SB-WB012, Share-bio, Shanghai, China).

### 4.8. RNA Extraction and Quantitative Real-Time PCR

Total RNA was extracted by using an RNA isolation kit (DP430, Tiangen Biotechnology Co., Ltd., Beijing, China). For reverse transcription, Hifair III reverse transcriptase (11141ES60, Yeasen Biotechnology Co., Ltd., Shanghai, China) was used. The mixture was prepared using the Hieff qPCR SYBR green master mix (11201ES08, Yeasen Biotechnology Co., Ltd., Shanghai, China), and the quantitative real-time PCR was performed using the LightCycler 480 system (Roche, Switzerland). β-actin was used as the endogenous references. The results were calculated using the log10 (2^−^^△△CT^) method. The primers used were as follows: forward primer (5′-GATGTGGCTGCTGTCAATG-3′) and reverse primer (5′-CACGCTGGGACGAAGG-3′) for EMC10; forward primer (5′-GAGTTTCCTAAAGCCATATTCTG-3′) and reverse primer (5′-ACACCACTACATTCTTCAAGC-3′) for ATP1B3; forward primer (5′-CGTAAAGACCTCTATGCCAACAC-3′) and reverse primer (5′-ACTCATCGTACTCCTGCTTGC-3′) for β-actin.

### 4.9. CoroNa Green Dye as Sodium Indicator

CoroNa Green dye is an improved green fluorescent sodium indicator that exhibits increased fluorescent emission upon binding to Na^+^, and the fluorescence intensity of cells loaded with the CoroNa Green dye can be detected. Briefly, the cells were digested using ethylenediaminetetraacetic acid-free enzymes and then gently resuspended in Hanks’ balanced salt solution (HBSS). CoroNa Green (C36676, Invitrogen, Carlsbad, CA, USA), a sodium-sensitive fluorescent dye, was dissolved into anhydrous dimethyl sulfoxide and diluted to a final concentration of 5–10 μM in cell suspension. Next, the mixture was incubated at 37 ℃ for 30 min, after which the loaded cells were washed twice with HBSS before fluorescence measurement. The average fluorescence intensity values of 10,000 cells were obtained using the Accuri^TM^ C6 flow cytometer (BD Biosciences, San Diego, CA, USA) with a fluorescein isothiocyanate (FITC) channel. Cytoplasm sodium was indicated by the fluorescence intensity values.

### 4.10. Immunohistochemistry

For immunohistochemical analysis, mouse testis and epididymis tissues were fixed in 4% (*w*/*v*) paraformaldehyde (PFA) at 4 °C overnight. Thereafter, tissues were dehydrated using a graded ethanol series, paraffin-embedded, and sectioned to approximately 5 μm. Next, the sections were de-waxed, rehydrated, subjected to antigen retrieval, and blocked with 10% donkey serum, followed by incubation with EMC10 (1:100) or ATP1B3 (1:100; ab231671, Abcam, Cambridge, MA, USA) antibody at 4 ℃ overnight. For colorimetric detection, the slices were thrice washed with 1× phosphate-buffered saline with Tween (PBST) and incubated with secondary antibody against rabbit (1:50; A0208, Beyotime, Shanghai, China) at 37 °C for 1 h. Subsequently, images were acquired under a BX-51 microscope (Olympus, Tokyo, Japan).

### 4.11. Immunofluorescence Staining

Spermatozoa were collected from the caput, corpus, and cauda epididymis of adult mice. Immunofluorescence was enhanced as previously described [36]. Briefly, sperm cells on slides were fixed with 4% PFA, incubated with 0.5% Triton X-100, and blocked with 10% goat serum (AR1009, Boster Biological Technology Co., Ltd., Wuhan, China). The slides were then incubated with antibody against ATP1B3 (1:100; ab231671, Abcam, Cambridge, MA, USA) at 4 °C overnight, and anti-rabbit Alexa Fluor 568 (1:1000; ab175471, Abcam, Cambridge, MA, USA) was used as the secondary antibody for incubation at 37 °C for 1 h. Peanut agglutinin (PNA) is a lectin that binds specifically to the outer acrosome membrane of sperm [37]. Here, FITC-labeled PNA (PNA-FITC; 1:2000, L7381, Sigma-Aldrich, St. Louis, MO, USA) was used to stain the sperm acrosome and co-incubated with secondary antibodies. Finally, sperm nuclei were revealed using 4′,6-diamidino-2-phenylindole (DAPI; 1:1000, D9542, Sigma-Aldrich, St. Louis, MO, USA) stain at 37 °C for 10 min. Images were acquired with confocal laser scanning microscopy (Leica, Heidelberg, Germany).

### 4.12. Statistical Analysis

Image-J software version1.8.0 (National Institutes of Health, Bethesda, Maryland, USA) was used to measure the gray values of each band in the Western blots. Data analyses were performed using SPSS software version 23 (IBM Corp., Armonk, NY, USA). Results corresponding to continuous variables are summarized as mean ± standard deviation, while those corresponding to categorical variables are presented as percentages. To compare the mean values of two groups, Student’s *t*-test was performed. Furthermore, after scEMC10 quartile grouping, Kruskal–Wallis and chi-squared tests were performed to analyze continuous and categorical variables, respectively. Spearman rank correlation analysis was also performed to determine the relationships between scEMC10 in seminal plasma and sperm motion parameters. Two-sided *p*-values < 0.05 indicated statistical significance.

## Figures and Tables

**Figure 1 ijms-23-10069-f001:**
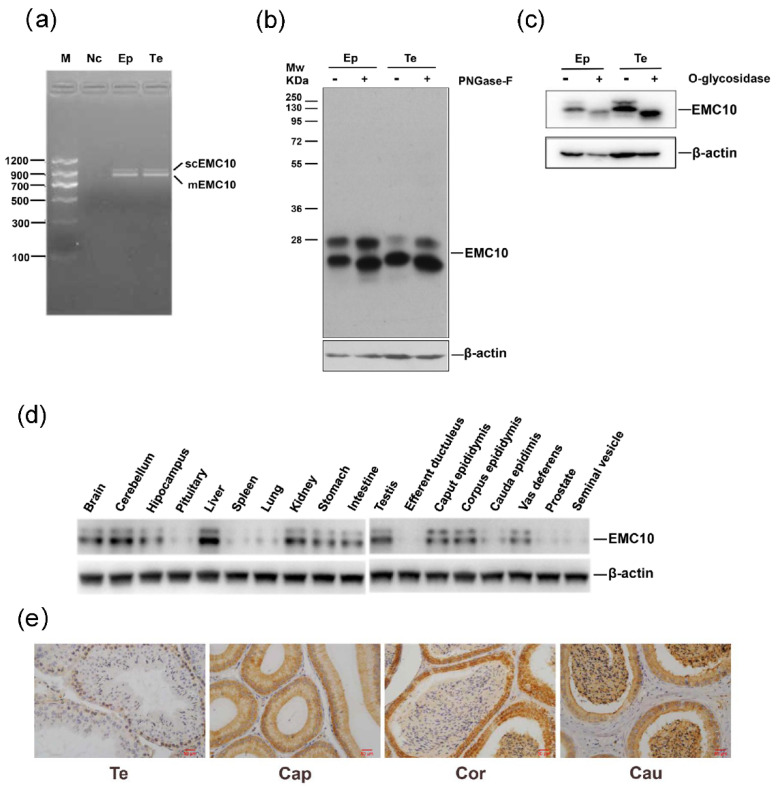
Characterization of EMC10 in mouse. (**a**) Two isoforms of *Emc10* gene were amplified in the testis and epididymis tissues of mouse by RT-PCR and subject to agarose gel electrophoresis. (**b**,**c**) Protein lysate was treated with *N*-glycosidase-F and *O*-glycosidase, and then immunoblotted with EMC10 antibody. (**d**) EMC10 protein profiling in mouse. (**e**) Immunohistochemistry analysis of EMC10 in the testis and epididymis tissues of mouse. Scale bar = 50 μm. M, marker; Nc, negative control; Te, testis; Ep, epididymis; Cap, caput epididymis; Cor, corpus epididymis; Cau, cauda epididymis; PNGase-F, *N*-glycosidase F.

**Figure 2 ijms-23-10069-f002:**
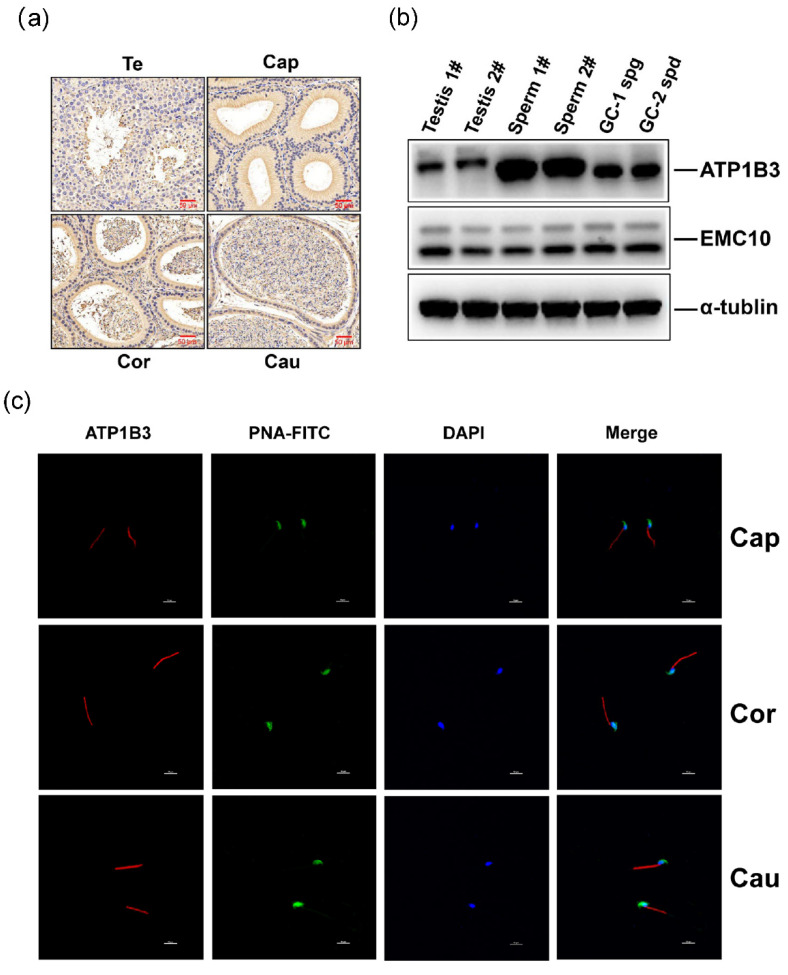
ATP1B3 expression and localization in male germ cells of mouse. (**a**) Immunohistochemistry analysis of ATP1B3 in the testis and epididymis tissues of mouse. Scale bar = 50 μm. (**b**) EMC10 and ATP1B3 protein expression measured in the testis, spermatozoa, GC-1, and GC-2 cells by Western blotting. (**c**) Immunofluorescence analysis of ATP1B3 localization (red) in spermatozoa from mouse epididymis (100×). Scale bar = 10 μm. Te, testis; Cap, caput epididymis; Cor, corpus epididymis; Cau, cauda epididymis; PNA-FITC (green), sperm acrosome; DAPI (blue), sperm nucleus.

**Figure 3 ijms-23-10069-f003:**
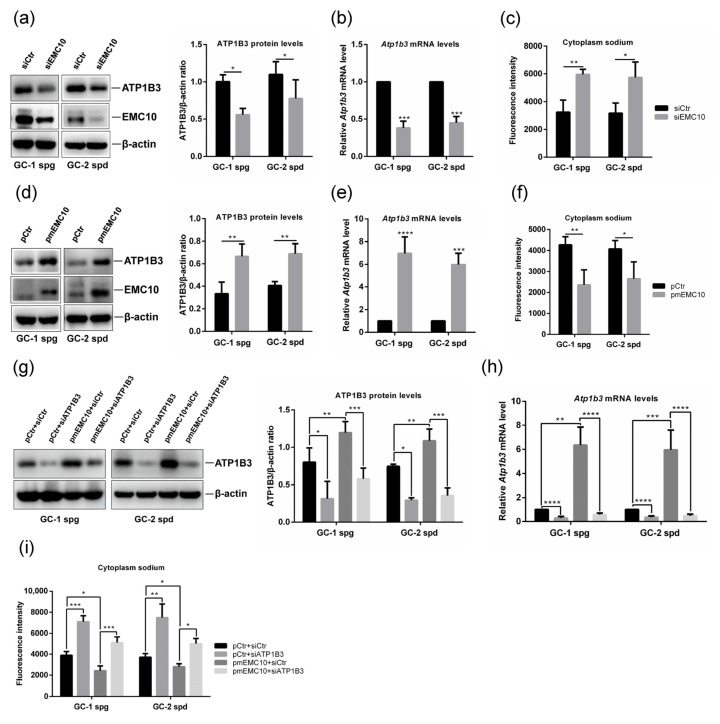
mEMC10 maintains the homeostasis of cytoplasm sodium via positively regulating ATP1B3 in germ cells. (**a**) The protein levels of EMC10 and ATP1B3 in GC-1 and GC-2 cells after the transfection of siRNA targeting mouse *Emc10* gene or control siRNA for 48 h. (**b**) *Atp1b3* mRNA levels in GC-1 and GC-2 cells transfected with the indicated siRNAs for 48 h. (**c**) Cytoplasm sodium was determined using CoroNa Green, a sodium-sensitive fluorescent dye, in *Emc10* siRNA or control siRNA-transfected GC-1 and GC-2 cells. (**d**) mEMC10 and ATP1B3 protein expression, (**e**) *Atp1b3* gene expression, and (**f**) cytoplasm sodium in GC-1 and GC-2 cells transfected with a plasmid-encoding mouse membrane-bound EMC10 or control plasmid for 48 h. The levels of (**g**) ATP1B3 protein, (**h**) *Atp1b3* mRNA, and (**i**) cytoplasm sodium in GC-1 and GC-2 cells transfected with different plasmids and siRNAs as indicated for 48 h. siCtr, control siRNA; siEMC10, EMC10 siRNA; pCtr, control plasmid; pmEMC10, mEMC10 plasmid. * *p <* 0.05, ** *p <* 0.01, *** *p* < 0.001, **** *p* < 0.0001.

**Figure 4 ijms-23-10069-f004:**
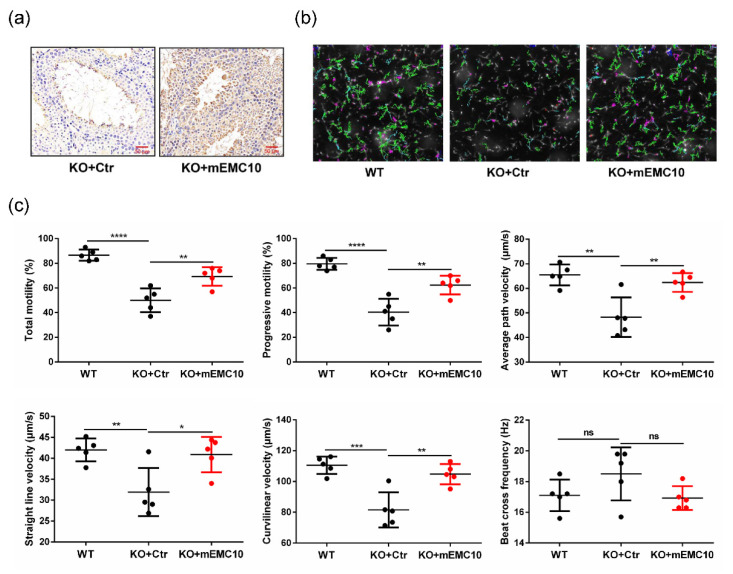
Intra-testis overexpression of mEMC10 rescues sperm motility in *Emc10* knockout (KO) mouse. (**a**) Immunohistochemistry staining of mEMC10 in the testes of *Emc10* KO mice after intra-testis injection of lentivirus expressing mEMC10 or control lentivirus for one month. Scale bar = 50 μm. (**b**) Representative images of sperm movement trajectories of wild-type and *Emc10* KO mice with intra-testis injection of mEMC10 lentivirus or control lentivirus under a 4× objective lens. Green trajectories represent the trajectories of rapid-moving spermatozoa. (**c**) Sperm motility was analyzed in spermatozoa of wild-type and *Emc10* KO mice with intra-testis injection of mEMC10 lentivirus or control lentivirus by computer-assisted sperm analysis. KO, *Emc10* knockout; WT, wild-type; Ctr, control lentivirus; mEMC10, lentivirus expressing mEMC10; ns, not significant. * *p <* 0.05, ** *p <* 0.01, *** *p <* 0.001, **** *p <* 0.0001.

**Figure 5 ijms-23-10069-f005:**
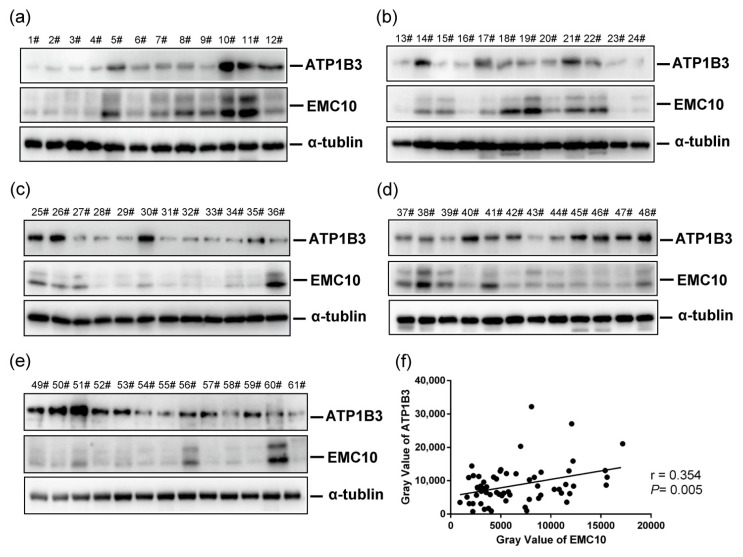
The protein levels of EMC10 and ATP1B3 in human spermatozoa. (**a**–**e**) EMC10 and ATP1B3 protein expression were determined by Western blotting in 61 human sperm samples. (**f**) Based on the Western blotting results shown in (**a**–**e**), gray values of EMC10 and ATP1B3 were evaluated using ImageJ software version1.8.0 (National Institutes of Health, Bethesda, MD, USA). Correlation analysis of the levels of EMC10 and ATP1B3 protein in these samples were performed using Spearman’s analysis.

**Figure 6 ijms-23-10069-f006:**
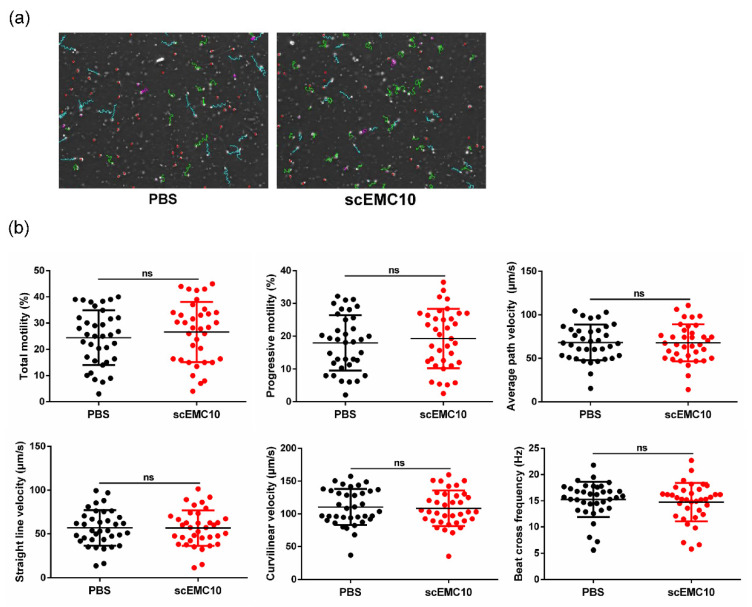
scEMC10 cannot improve sperm motility in humans with asthenozoospermia. (**a**) Representative images of movement trajectories of scEMC10- or PBS-treated spermatozoa under a 10× objective lens. Green trajectories represent the trajectories of rapid-moving spermatozoa. (**b**) Sperm motility analysis of those spermatozoa by computer-assisted sperm analysis. ns, not significant. The sperm samples were incubated with 1 μg/mL recombinant scEMC10 protein or phosphate-buffered saline (PBS) at 37 °C for 1 h.

**Table 1 ijms-23-10069-t001:** Characteristic parameters of sperm samples in a cohort of Chinese participants.

	Normal-Motile Spermatozoa	Low-Motile Spermatozoa	*p*
Number (*n*)	122	67	
Age (years)	32.8 ± 6.7	32.7 ± 7.3	0.961
Semen volume (mL)	2.70 ± 1.40	3.08 ± 1.54	0.099
Sperm count (×10^6^/mL)	194.76 ± 160.17	140.37 ± 148.79	0.021
Total motility (%)	58.10 ± 11.31	25.15 ± 10.52	<0.0001
PR (%)	55.30 ± 11.47	22.18 ± 10.65	<0.0001
VAP (μm/s)	46.65 ± 10.07	38.05 ± 14.25	<0.0001
VCL (μm/s)	79.23 ± 15.91	64.60 ± 22.05	<0.0001
VSL (μm/s)	39.02 ± 9.50	31.76 ± 13.13	<0.0001
BCF (Hz)	8.88 ± 3.17	7.49 ± 3.83	0.014
scEMC10 (ng/mL)	6.95 ± 3.50	7.12 ± 2.73	0.706

Spermatozoa with total motility ≥40% and PR ≥ 32% were considered as normal-motile spermatozoa and those with total motility <40% or PR < 32% were considered as low-motile spermatozoa (asthenospermia). Continuous variables are summarized as mean ± standard deviation. Student’s *t*-test was used to compare the mean values of the two groups. PR, progressive motility; VAP, average path velocity; VSL, straight line velocity; VCL, curvilinear velocity; BCF, beat cross frequency.

**Table 2 ijms-23-10069-t002:** Characteristic parameters of sperm samples grouped into seminal plasma scEMC10 quartiles.

	Quartile 1	Quartile 2	Quartile 3	Quartile 4	*p*
scEMC10 (ng/mL)	0.83–4.14	4.17–6.08	6.12–8.16	8.17–17.56	
Number (*n*)	47	50	45	47	
Age (years)	32.4 ± 5.9	32.6 ± 5.5	30.6 ± 6.4	35.2 ± 8.9	0.027
Semen volume (mL)	2.5 ± 1.4	3.2 ± 1.5	2.8 ± 1.3	2.8 ± 1.6	0.123
Sperm count (×10^6^/mL)	185.7 ± 179.1	213.4 ± 156.9	151.5 ± 131.0	147.8 ± 155.9	0.064
Total motility (%)	48.4 ± 22.5	50.5 ± 19.0	42.3 ± 17.2	44.0 ± 17.3	0.101
PR (%)	45.4 ± 22.3	47.2 ± 19.9	39.9 ± 17.0	41.3 ± 17.5	0.149
VAP (μm/s)	43.4 ± 11.7	43.2 ± 12.1	44.7 ± 10.5	43.2 ± 15.1	0.95
VCL (μm/s)	75.1 ± 17.2	75.0 ± 20.1	75.3 ± 17.2	70.9 ± 23.2	0.659
VSL (μm/s)	35.3 ± 11.4	36.0 ± 10.8	37.7 ± 9.2	36.8 ± 14.0	0.659
BCF (Hz)	7.8 ± 3.4	8.8 ± 3.6	8.3 ± 3.0	8.7 ± 3.8	0.651
Asthenospermia (%)	29.8%	30.0%	48.9%	34%	0.367
Male infertility (%)	21.2%	18%	20%	19.1%	0.981

Asthenospermia and male infertility were defined according to the World Health Organization (WHO). Kruskal–Wallis and chi-squared tests were used to analyze continuous and categorical variables, respectively. PR, progressive motility; VAP, average path velocity; VSL, straight line velocity; VCL, curvilinear velocity; BCF, beat cross frequency. Continuous variables and categorical variables are summarized as mean ± standard deviation and number for proportion.

**Table 3 ijms-23-10069-t003:** Spearman’s correlation analysis between seminal plasma scEMC10 levels and various parameters of sperm motility.

	scEMC10 Level	*p*
Sperm count (×10^6^/mL)	−0.106	0.148
Sperm motility (%)	−0.142	0.052
PR (%)	−0.129	0.076
VAP (μm/s)	−0.011	0.879
VCL (μm/s)	−0.134	0.066
VSL (μm/s)	0.044	0.545
BCF (Hz)	0.012	0.87

Data are represented as Spearman’s correlation coefficients. PR, progressive motility; VAP, average path velocity; VSL, straight line velocity; VCL, curvilinear velocity; BCF, beat cross frequency.

## Data Availability

Not applicable.

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
