# Peer review of "Membrane-Bound EMC10 Is Required for Sperm Motility via Maintaining the Homeostasis of Cytoplasm Sodium in Sperm"

_ijms, 2022, doi:10.3390/ijms231710069_

Round 1

Reviewer 1 Report

The paper entitled: “Membrane-bound EMC10 is required for sperm motility via 2 maintaining the homeostasis of cytoplasm sodium in sperm” by Liu and coauthors, describes importance of EMC10 isoform m for sperm motility including sodium homeostasis compared to other isoform m. This paper is interesting. It uses mouse model as well as humans samples. However it contains many weaknesses that should be eliminated before this paper could be accepted for publication in this Journal. Below I give some examples of weak points that should be corrected. There are many others and I suggest the Authors to  read the paper entirely to improve it.

Criticisms:

1.      I would remove the fragment starting in line 24 the sentence “Clinically, there was no correlation between seminal plasma scEMC10 levels and sperm motility” – it’s not crucial for the abstract. I would keep “There existed a positive association between ATP1B3 and EMC10 protein levels in human spermatozoa”. After that I would include the result about sodium measurement to justify the next sentence - “This highlights the important roles of the membrane-bound…”.

2.      Introduction requires profound remodeling. Introduction starts with infertility which is OK. But after that there is a long fragment (line 50 – 62 in previous studies… “ about completely unrelated to infertility roles of that protein and it cut the flow. This should be removed since there is too much about the role of EMC10 unrelated to infertility. In the second part of Introduction fragment between line 75- 90 should be removed. The aim and main results should be shortly described at the end of Introduction.

3.      The mouse cell lines GC-1 and GC-2 are not explained. We learn from Materials and Methods what it is but this part is at the end of the article. These lines should be descried in the Results part.

4.      Line 141-142 : The spatial overlap of ATP1B3 and EMC10 in the middle piece of spermatozoa implies an interaction between them. – This sentence is to strong such classical colocalization as it was done in this paper is not precise enough to discuss interaction. For such purpose such technics as FRET or biochemical – co-immunoprecipitation is necessary. ”Imply” is slightly to strong here. It could be suggested instead that more precise methods should be used to conclude about potential interaction.

5.      Fig.  1b better indicate that both bands represent EMC10. Fig. 1d. In panel Te what is described in the text is not seen. Should make a bigger picture and indicates all the stages mentioned in line: 118-119 by arrows. Alternatively, you can use insets.

6.      It is not explained anywhere what is PNA-FITC (Fig. 2c). I did not find it in Mat and Meth.

7.      Omit the first sentence in 2.1 (line 93) since starting with human while mouse is described here is confusing.

8.      In 2.2 chapter add “mouse germ cells”

9.      Line 110: “We observed N-glycosidase F treatment slightly decreases the sizes of the both bands in the gel” – it is not shown, why?

10.   Fig. 2a what is in the text is not seen in Fig. Use bigger magnification or use insets. Fig. 2c – bad quality use better quality nothing is seen here.

11.   Gc-1 and Gc-2 not described at all. PNA- what is that?

12.   Fig. 3, Fig. 4C, Fig. 6B and Fig. 5 (number over the blots indicating patients).  Fonts are too small, impossible to read in paper version. This should be definitely corrected.

13.   No one fluorescence result is visible.

14.   14. While talking about sodium measurements result (even if details are described in Methods section) the principle should be shortly explained.

15.   Line 230 “from” instead of “form”.

16.   Line 169 should be “regulating”

17.   β-actin as well as  α-tubulin are usually too much exposed and its difficult to compare the protein amount in each slot. Shorter exposure should be placed instead.

18.   Line 232-234 “Correlative analysis indicated the levels of ATP1B3 are positively associated with those of EMC10 in human spermatozoa (Fig. 5f). This finding accords with the observation that mEMC10 positively regulates ATP1B3 in germ cells (Fig. 3).” This sentence is too strong since the Authors underlined lack of antibody specificity for mEMC10.

19.   From Fig 5a-e it’s not clear which individuals are patients and which are healthy. If there are healthy all of them differences of ATP1B3 and EMC10 content are not essential for health. That should be very well explained.

Reviewer 2 Report

The present manuscript characterizes two isoforms of EMC10 in mouse testis and epididymis, and described that specifically the mEMC10 isoform is related to the cytoplasm sodium homeostasis by positively regulating ATP1B3 expression in germ cells, which is associated with sperm motility. Furthermore, the function of EMC10 also is tested in human spermatozoa which could be helpful to establish the bases of the understanding the male infertility and could be interesting as therapeutical potential. However, few clarifications and modification are required before further consideration.

Questions and concerns to be addressed properly:

1- All the text:

-  Moderate English editing should be reconsider.

- Acronyms should be explained when they first appear. Author should revise this aspect on the whole manuscript. 

2-Introduction:

-      On the first paragraphs, authors should reorganize the different ideas corresponding to mouse and human information about both subunits of EMC10, to improve the understanding of the section.

-         In the last paragraph of this section, the aim needs to be reformulated, being more direct and in present time, not in future. Maybe there is no need to describe all the steps that are going to be developed during the manuscript, as it sounds more like abstract section.

-         Authors should remove the last sentences of the final part of the section (from line 81 to line 90), as they are only the guidelines of what it should be described in the introduction section.

3-Results:

-          Most of the images are really small, and it is impossible to appreciate what authors want to show. In my opinion those figures should be amplified and restructured to facilitate the reading of the figure text.

-          In figure 1, there are two parts of section B (WB). In my opinion it would be easier to follow the text, if the section B is separated in two sections (B and C), and if the other sections of the figure 1 are renamed to continue with the listing.

-          In most figures, the b-actin or the a-tubulin are very irregular, and it makes difficult to compare the results and conclude the same ideas that are described on the text. This is the case of figure 1 b and c; figure 2 a, d and g; and figure 5. Authors should choose WB images were the normalizer is better.

-          In my opinion, authors should include densitometry of WB experiments in all the figures (also in 1, 2 and 5) and not only in figure 3.

-          Regarding figure 2, the immunofluorescence pictures are not very clear. They are too small and not with an optimized quality to see the differences explained in the text. Furthermore, the red signal is too saturated and it makes the background also to look red. Authors should improved all of that and add negative control pictures for each secondary antibody, at least in supplementary data. Also it would enrich the result to add the information of the optic amplification (X) and a better scale bar.

-          In both results of the blue trajectories of motile spermatozoa (figure 4b and 6a) the described differences are not appreciable. Authors should improve the image.

-          Reading the results, I miss an explanation of why the authors use lentivirus for mEMC10 subunit and Adenovirus for scEMC10 subunit.

-          In section 2.4 authors explain that the antibody used to immunoblot EMC10 could not differentiate mEMC10 from scEMC10. Is there any possibility to find specific antibody for at least one of both subunits to test the correlation of the different EMC10 isoforms with ATP1B3 in human spermatozoa? In my opinion, this extra experiment should enrich the manuscript and make it more suitable to publish.

-          Is it possible to also associate mEMC10 levels with sperm motility and other parameters in human spermatozoa as it is done for scEMC10 levels (Section 2.5)? And mEMC10 levels with asthenospermic sperm motility parameters (section 2.6)? In my opinion, these extra experiments should enrich the manuscript and make it more suitable to publish. 

4-Discussion

-          In the most part, discussion reads more like an extension of Results. This section should be enriched by comparing the obtained results with similar works and reorganizing it to look more structured.

-          Furthermore, for the first paragraph, always it is recommended to introduce again the aim of the publication, with a brief explanation of the main results, to give rise to be able to start discussing each result comparing it with the available bibliography.

5-Materials and methods

-          In all the section, authors should add the commercial house and the reference of each used products, reagents or machines, as the manuscript should have enough information for the readers to reproduce all the experiments. On the other hand, if the product or reagent is self-done, the grs or % of the used products need to be detailed.

-    Authors refer most of the times to some procedures carried out in other publication. In my opinion, they should add a brief description of those procedures to facilitate the understanding of them.

-     In section 4.4, authors should clarify first if the used cells are bought (and if so, their commercial house and the reference), or if they are obtained following a special procedure. Furthermore, they should described the characteristics and differences of both male germ cell lines GC-1 spg and GC-2 spd. The same should be explained for 293T cells. If I have not read it wrong, this last cells are not mentioned in the rest of the manuscript, so why are they described in the materials and methods section?

- All the times that antibodies are used (for immunofluorescence, immunohistochemistry or WB), the antibodies commercial house, reference, and used dilution should be added, both for primary antibodies and for secondary antibodies.

6-References

- Maybe the reference section is too reduced. More references shoud be added to enrich the discussion section.

- Furthermore, there is an excess of self-citation with that 21 reference.

Reviewer 3 Report

The main question in this work is an attempt to distinguish which isoform of endoplasmic reticulum membrane protein complex subunit 10 (EMC10): membrane-bound (mEMC10), or secreted (scEMC10) supports sperm motility by regulating the sodium level inside the spermatozoa. The authors checked also the therapeutic effects of both these isoform on of Emc10 knockedout spermatozoa, their motility and fertility.  

The conducted research is innovative and revealing. I consider that the topic of this paper is original and relevant in the field. The conducted research is innovative and revealing.

This work represents the first effort to explore the impacts of the distinct isoforms of EMC10 on a certain biological activity, and reveals that mEMC10, rather than scEMC10, governs the adequate intracellular sodium concentration in sperm. Based upon this study, the authors propose a novel mechanism of the regulation of cytoplasm sodium and motility of sperm, which can serve as a new therapeutic way in treatment of asthenospermia.

I believe that the methodology is very comprehensive and very well structured. The descriptions of the materials and methods are very clear and good I see that the authors put a lot of effort and research resources into carrying out this research.

The conclusions are in line with the evidence and arguments presented and relate well to the main questions posed.

The references presented are adequate, they were selected with care and knowledge of the subject. The references are very current and well chosen.

The presentation of the results in the form of figures and tables is very good.

On lines 81 to 90, is there any mistake?

From line 215 to line 225, Fig. S3a, Fig. S3b, Fig. S3c, Fig. S4a and S4b, where are these tables?

Round 2

Reviewer 2 Report

I appreciate the efforts of the authors to be able to satisfy the changes proposed by the reviewers in such a short time. However, a few more changes and questions are proposed before further considerations:

Questions and concerns to be addressed properly:

Introduction:

-          On the first paragraphs, authors should reorganize the different ideas corresponding to mouse and human information, it is a little bit messy.

Results:

-          In figure 1, there are two parts of section B (WB). In my opinion it would be easier to follow the text, if the section B is separated in two sections (B and C), and if the other sections of the figure 1 are renamed to continue with the listing.

-          In most figures, the b-actin or the a-tubulin are very irregular, and it makes difficult to compare the results and conclude the same ideas that are described on the text. This is the case of figure 1 b and c; figure 2 a, d and g; and figure 5. Authors should choose WB images were the normalizer is better.

-          Regarding figure 2, the immunofluorescence pictures have not an optimized quality to appreciate the differences explained in the text. Authors should improved the picture quality and add the information of the optic amplification in the description of the image.

-          In both results of the blue trajectories of motile spermatozoa (figure 4b and 6a) the described differences are not appreciable. Authors should improve the image.

-          In their response to my suggestion of associating mEMC10 levels with normal and asthenospermic sperm motility and others parameters, the authors have response that for the diagnostic evaluation they were limited by the lack of specific antibodies for EMC10 isoforms. But if I am not wrong 2.5 and 2.6 sections are carried out with patients sperm parameters and a correlation with scEMC10 isoform levels.  If authors are limited by de lack of specific antibodies for both isoforms how can they measure scEMC10 levels? In my opinion, the addition of the information of mEMC10 levels should enrich the manuscript and make it more suitable to publish.

Discussion

-          In the most part, discussion reads more like an extension of Results. This section should be better structured and enriched.

Materials and methods

-          In all the section, authors should add not only the commercial house, but also the reference of each used products, reagents or machines, as the manuscript should have enough information for the readers to reproduce all the experiments.

-          In section 4.4, authors should clarify first if the used cells are bought (and if so, their commercial house and the reference), or if they are obtained following a special procedure. Furthermore, they should described the characteristics and differences of both male germ cell lines GC-1 spg and GC-2 spd, which specific cells are they? Which important biological characteristic give to the results?

Round 3

Reviewer 2 Report

I appreciate the efforts of the author to be able to satisfy the changes proposed by the reviewers in such a short time. Now I think the manuscript could be consider to be accepted in the present form as the discussion and most of the result figures have improved importantly.

Author Response

We appreciated for your helpful suggestion, and were delighted to have reached your satisfaction.